# Imaging in Upper Tract Urothelial Carcinoma: A Review

**DOI:** 10.3390/cancers15205040

**Published:** 2023-10-18

**Authors:** Lucas A. Tsikitas, Michelle Diamond Hopstone, Alex Raman, Vinay Duddalwar

**Affiliations:** Department of Radiology, Keck School of Medicine, University of Southern California, Los Angeles, CA 90033, USA; lucas.tsikitas@med.usc.edu (L.A.T.); michelle.hopstone@med.usc.edu (M.D.H.); alex.raman@med.usc.edu (A.R.)

**Keywords:** upper tract urothelial carcinoma, imaging

## Abstract

**Simple Summary:**

Urothelial carcinoma, a cancer of the urinary tract, is relatively common in the urinary bladder, termed the lower urinary tract. However, it is much less common in the upper urinary tract, which consists of the pelvicalyceal system and ureters. Medical imaging plays an important role in detection, diagnosis, and treatment planning of this uncommon disease. We aim to review the imaging methods currently available and future directions in the field of radiology to aid clinicians in treatment planning.

**Abstract:**

Medical imaging is a critical tool in the detection, staging, and treatment planning of upper urinary tract urothelial carcinoma (UTUC). This article reviews the strengths and weaknesses of the different imaging techniques and modalities available clinically. This includes multidetector computed tomography (CT), multiparametric magnetic resonance imaging (MRI), ultrasound (US), and positron emission tomography (PET) for the detection, staging, and management of UTUC. In addition, we review the imaging techniques that are being developed and are on the horizon but have not yet made it to clinical practice. Firstly, we review the imaging findings of primary UTUC and the techniques across multiple modalities. We then discuss imaging findings of metastatic disease. Lastly, we describe the role of imaging in the surveillance after resection of primary UTUC based upon current guidelines.

## 1. Introduction

UTUC is a type of cancer that arises from the urothelial cells lining the renal pelvicalyceal system and ureters. It is far less common than lower tract urothelial carcinoma, making up for only 5–7% of all urothelial carcinomas [1]. Detection is usually incidental or in the clinical setting of hematuria or flank pain [2]. UTUC is most often detected in the renal pelvis [3]. It is then most commonly detected in the distal third of the ureter, then mid ureter, and lastly proximal ureter occurring at rates of 73%, 24%, and 3%, respectively [3]. As UTUC is much rarer than lower urinary tract urothelial carcinoma, there are limited epidemiologic data; the overall incidence of UTUC is reported as 1–2 cases per 100,000 people [1,4,5]. Approximately 11–13% of patients with UTUC develop metachronous UTUC tumors, underscoring the importance of optimized imaging techniques in primary detection and surveillance [1,6]. Treatment of UTUC widely varies depending on the location and number of masses, presence of metastatic disease, and whether the patient is high- or low-risk. Surgical management ranges from kidney-sparing resection or chemoablation in localized low-risk disease to open radical ureteronephrectomy with lymph node dissection in localized high-risk disease [1]. Additionally, patients over the age of 70 who undergo radial nephrectomy may have worse outcomes than those who have noninvasive treatment [7]. As such, detection of the full disease extent is crucial prior to developing the clinical treatment plan.

## 2. Detection and Diagnosis

UTUC typically presents on imaging in one of three ways: a filling defect within the renal pelvicalyceal system or ureter; focal thickening of a segment of urothelial lining, often with prominent focal enhancement; or as an infiltrative mass [2,3]. Of note, one general feature of an infiltrative renal urothelial carcinoma is that the contour of the involved kidney is preserved, helping to differentiate itself from RCC [8].

### 2.1. UTUC Staging

The staging of UC is based on the tumor, node, and metastasis (TNM) system, which considers the size and extent of the tumor, lymph node involvement, and presence of distant metastases [9]. The staging system for UTUC ranges from Tis (carcinoma in situ) to T4 (tumor invades adjacent organs or structures) [9].

If there is a fat plane or layer of excreted contrast that separates a pelvicalyceal mass from the normal renal parenchyma, the tumor can be classified as T1 or T2 [8]. Although desirable for treatment planning, differentiation between T1 and T2 on imaging is difficult. A T3 lesion will lose this fat plane or layer of contrast and may show enhancement in the adjacent renal parenchyma [8]. Invasion into the renal parenchymal is designated as T4 disease [8].

In a study of 188 patients with UTUC treated using radical nephroureterectomy, distant metastatic disease was found to occur most commonly in multiple organs sites, in 30% of cases [6]. Single-organ metastatic disease was then highest in the lungs at 28% of cases, followed by both liver and bone at 13%, and distant lymph nodes at 10% [6]. Lung metastases typically present as multiple pulmonary nodules [10]. Cystic lung nodules have also been reported, although less commonly [11]. Metastatic disease in the liver is typically seen as multiple hypoattenuating masses, although solitary masses are reported in approximately 10% of cases [10]. Osseous metastatic disease can present as sclerotic, lytic, or mixed picture masses, only rarely resulting in vertebral body compression fractures or spinal cord compression [10]. Metastatic lymph nodes are often enlarged and bulky conglomerates in various locations based on where the primary tumor originated. Regional lymph nodes for primary intrarenal and proximal ureteral UTUC occur in the perihilar and retroperitoneal stations. For the distal ureters, regional lymph nodes include the hypogastric, obturator, iliac (internal, external), perivesical, pelvic (not otherwise specified), sacral, and presacral lymph node stations [6,10]. Common iliac nodes are not included as regional nodes and are considered distant metastases (M1). Single lymph node metastases are associated with better clinical outcomes when compared to an increased number of nodes or increased nodal density [6]. Less commonly, pleural, adrenal gland, brain, peritoneal, and bowel metastases have also been reported [10].

### 2.2. Computed Tomography

CT urography (CTU) is a technique that combines multiple CT acquisitions typically with and without contrast [2,3]. While UTUC is an uncommon local for urothelial carcinoma, the upper urinary tract is considered the second most common site of involvement after the bladder, and multifocality is a hallmark of the disease. Therefore, proper distention of the renal pelvis and ureters is essential for detection, and the optimization of CTU protocols is essential. Suboptimal technique can lead to poor urinary tract distention, making the detection of subtle tumors impossible. CTU has a sensitivity of over 90% in patients with painless hematuria and is accepted as routine evaluation for this clinical indication [2]. Meta-analysis has demonstrated the best diagnostic accuracy for UTUC of all non-invasive medical imaging with CTU [5]. Specifically, CT urography has been found to have a pooled sensitivity of 92% for the detection of UTUC (95% CI; 0.85–0.96), and pooled specificity of 95% (95% CI; 0.88–0.98) in a meta-analysis of 13 studies comprising 1233 patients [5]. Additionally, CTU has sufficient sensitivity for the detection of additional common causes of hematuria [2,3]. Below, we discuss the various CT acquisitions encountered in CTU.

Noncontrast CT has very limited to no utility for the identification of primary UTUC, as there is no natural contrast attenuation difference between the primary tumor and the normal renal collecting system and ureters. It is primarily useful in patients presenting with hematuria by elucidating hyperattenuating renal calculi. Occasionally, primary UTUC may show fine encrusted calcifications, which are difficult to differentiate from renal calculi with just noncontrast imaging [8]. Additionally, space-occupying lesions, if detected, are difficult to differentiate from other renal masses in a noncontrast-enhanced study.

Following the administration of intravenous contrast, few additional etiologies of hematuria can be evaluated (Figure 1). In the corticomedullary phase of contrast (approximately 25–30 s post injection), UTUC can be seen as an infiltrative mass with arterial hyperenhancement [1,2]. Small urothelial carcinomas in the ureters tend to demonstrate early arterial enhancement, which helps differentiate them from benign entities such as blood clots or sloughed papillae in papillary necrosis as well as having a more central intraluminal position rather than asymmetrically on the ureteral wall [1,2,3]. Renal cell carcinoma can also be identified using early arterial enhancement but tends to exhibit a mass effect and deform the normal renal contour [1,2,3].

The nephrographic phase of contrast allows for the detection of some urothelial carcinomas, as well as renal cell carcinomas [2,3]. The detection of primary pelvicalyceal UTUC remains limited, however, as the ureters are not well opacified and the renal collecting systems are isoattenuating to adjacent renal parenchyma. This makes soft tissue masses, flat epithelial lesions, or focal urothelial thickening extremely difficult to detect [1].

Imaging obtained during the excretory phase of contrast administration (approximately 4–8 min post injection) allows for optimal opacification and distention of the ureters, resulting in maximum opacification of the collecting system and ureters [2,3]. On excretory imaging, UTUC appears as ureteral filling defects or irregularities of the calyx or infundibular narrowing [2,3]. The addition of image reconstruction techniques such as excretory maximum-intensity projections (MIPs) and 3D reconstructions helps reveal UTUC in the intrarenal collecting systems as calyceal amputation or destruction can be easier to visualize [2]. A careful review of the imaging dataset using a combination of multiple viewing windows and reconstructions is critical for an accurate imaging review. The use of three-dimensional (3D) imaging techniques has also been found helpful, as it can make a conspicuous lesion stand out more than relying on the evaluation of axial images alone [2,3].

CTU is performed in one of two techniques, depending on the clinical scenario, institutional volume, and staffing: the single bolus technique and split bolus technique. The single bolus technique involves a noncontrast image acquisition followed by injection of a single contrast agent bolus, followed by multiple CT acquisitions during the corticomedullary, nephrogenic, and delayed excretory phases, leading to the highest sensitivity for all etiologies of hematuria. The delayed excretory phase of the single bolus technique provides excellent visualization of the urinary tract as the entire contrast bolus contributes to opacification/distension of the collection system [2,3]. Additionally, the single bolus technique is simpler and quicker to perform compared to the split bolus technique [2,3]. The largest downside of the single bolus technique is its undesirably high radiation dose [2,3].

The split bolus technique can include a noncontrast image acquisition, although this is sometimes omitted. During a split bolus exam, the same contrast volume is administered, but in two separately timed boluses, achieving a combined nephrographic and excretory phase, decreasing the number of image acquisitions and thus radiation dose [2,3]. While decreasing radiation dose, especially over a patient’s lifetime, this technique provides questionable visualization of the bladder and distal ureters, particularly for small UCs, as only half the volume of contrast contributes to collecting system opacification/distension [2,3]. Additionally, the lack of a corticomedullary phase decreases the sensitivity for detection of renal cell carcinomas and small flat epithelial lesions.

Several other imaging techniques aimed to enhance distention of the distal ureters include the administration of IV furosemide prior to the study, and the administration of IV or oral hydration. Both have been shown to increase distention of the distal ureters, and thus sensitivity for detection; however, the administration of IV furosemide has workflow implications including the need for nursing staff to place IVs and administer medications [2]. Additional techniques have been historically used such as using compression belts and scanning the patient in the prone position, but there is a lack of data to support their effectiveness.

Another emerging CT technique for the detection of UTUC is dual-energy CT (DECT). DECT takes advantage of the attenuation phenomenon, or the amount of X-ray energy that is attenuated by individual tissues at different X-ray energies, determined by the physical properties of the tissues such as atomic number and density. By collecting data at two different X-ray energies, it allows for the creation of iodine attenuation curves specific to each tissue. By using this technique, it is possible to subtract the attenuation solely caused by the iodinated contrast material, thus creating virtual noncontrast images. This technique is thus able to maintain a noncontrast image from a contrast-enhanced dataset, removing the need to acquire a noncontrast image, resulting in radiation dose reduction to the patient. Another useful DECT strategy is the creation of virtual monochrome images (VMI), which allow for higher contrast images using a reduced contrast bolus dose at the cost of increased image noise, which is reduced through mathematical algorithms. DECT additionally allows for an improved reduction in beam hardening artifacts, which is often present due to post-surgical hardware. Lastly, DECT allows for the evaluation of images using post-processing color-coded displays based upon the iodine uptake of tissues, greatly improving the detection of renal and urothelial lesions and differentiating solid and cystic renal masses. While DECT can greatly aid in the detection of UTUC, as a newer technology, it comes at a higher equipment cost and greater post-construction complexity, and little literature is currently available in the field of UTUC [12,13].

### 2.3. MRI

MRI urography (MRU) has been found to have a sensitivity of 75% after the administration of contrast for the detection of UTUC, but it has been shown to have equal specificity for UTUC detection compared to CTU [5,14,15]. MRU is limited in the detection of small nephrolithiasis and has decreased diagnostic accuracy in patients with ureteral stents or nephrostomy tubes [14]. Compared with CTU, MRU provides decreased spatial resolution and the susceptibility to motion artifacts from both patient movement and ureteral peristalsis. Additionally, the T2* effect seen with dense gadolinium can lower the sensitivity for detection in contrast-enhanced excretory phase sequences [8,16]. As CTU has a higher sensitivity, faster imaging times, and increased patient throughput, MRU is typically reserved for patients with contraindications to iodinated contrast administration or radiation [5]. 

MRU is composed of multiple sequences (Figure 2). T2-weighted sequences are performed at standard and very long echo times to allow for quick hydrographic images. This sequence allows for the detection of filling defects or truncation of the pelvicalyceal system, ureters, and bladder, without the need for IV contrast administration [5,16]. Similar to the CTU technique, oral or IV hydration, and IV furosemide administration help distend the ureters [16]. These sequences are somewhat limited, however, as the ureters may not be fully distended and are prone to motion artifacts from peristalsis [16].

Pre-contrast T1 sequences demonstrated T1 hypointense signals in tumor cells, and chemical shift imaging enables the evaluation of intravoxel fat. After the administration of IV gadolinium, T1-weighted dynamic post-contrast sequences are obtained in the corticomedullary, nephrographic, and excretory pyelographic phases, allowing for the detection of enhancing masses, urothelial linings, and filling defects [16]. Similar to CTU, papillary lesions and infiltrative disease demonstrate diffuse contrast enhancement, differentiating themselves from benign differential diagnoses such as sloughed papillae and blood clots [15].

Diffusion-weighted imaging demonstrates increased signals in tumors with decreased apparent diffusion coefficient (ADC) values, due to the greater restriction of water movement in tumor cells [17]. 

The modern MRU technique is generally composed of T2-weighted sequences at standard and very long echo times, T1-weighted dual-echo chemical shift sequences, fat-saturated pre- and post-contrast T1-weighted sequences, and diffusion-weighted imaging sequences. Alternatively, MRU can be obtained without the administration of intravenous contrast in certain patient populations, like those who are pregnant or have chronic renal failure, at a significantly decreased sensitivity [18]. 

### 2.4. US/CEUS

Although less sensitive than CTU and MRU in identifying upper tract urothelial carcinoma, ultrasound remains a useful tool in the detection of UTUC (Figure 3), especially in cases of renal failure, contrast allergies, or in the setting of limited medical resources. 

On grayscale ultrasound, pelvicalyceal tumors are solid masses that can appear hypo, iso, or hyperechoic to the adjacent echogenic renal sinus fat and may occur with or without the associated hydronephrosis. Ureteral lesions are seen as intraluminal soft-tissue masses with or without hydroureteronephrosis. Small, nonobstructive tumors may be difficult to differentiate from the renal sinus fat in the absence of the associated hydronephrosis. Large, diffuse infiltrative tumors may also be difficult to distinguish from adjacent renal parenchyma on grayscale ultrasound. Color Doppler ultrasound may show vascular flow; however, upper tract tumors frequently show low or no-flow and may be difficult to distinguish from clot or debris [19,20,21]. 

Contrast-enhanced ultrasound is emerging as a promising technique to aid in the diagnosis of upper tract tumors. As of this writing, no clear sonographic enhancement pattern has been established for UTUC, with variable enhancement characteristics likely dependent on tumor grade [19,21]. Nevertheless, multiple studies have found that many urothelial tumors demonstrate early washout relative to the renal cortex [19,20]. Contrast-enhanced ultrasound can accurately differentiate solid, enhancing tumors from non-solid, non-enhancing material such as blood clot, debris, or pus. As a result, CEUS appears to be more accurate in estimating tumor size when compared to CTU/MRU and grayscale ultrasound, which can overestimate tumor size by 15–20% and 25%, respectively [21]. Conducted in real-time, CEUS is more sensitive than CTU in the detection of tumor microvascularization [19]. 

Limitations remain, however, as contrast-enhanced ultrasound cannot distinguish UTUC from other, less common types of upper tract tumors including epidermoid tumors, adenocarcinoma, or lymphoma [19]. Ultrasound evaluation of the ureter remains limited, as the ureter cannot be seen in its entirety.

### 2.5. PET/CT

18F-fluorodeoxyglucose positron emission tomography with computed tomography (FDG-PET/CT) is commonly used in cancer staging; however, its use in the evaluation of primary tumors of the urinary tract in general is limited due to physiologic excretion of the radiotracer through the urinary system. Nevertheless, multiple studies have shown that FDG-PET/CT has high diagnostic accuracy in the detection of lymph nodes and distant metastases during initial staging and restaging of urothelial carcinoma (Figure 4) [22,23,24]. 

11C-choline is a PET radiotracer with very late urinary excretion, allowing the urinary tract to be free from urinary radioactivity at the time of image acquisition [25]. Multiple small series studies have shown that urothelial carcinoma demonstrates 11C-choline uptake [25,26]. The results of these studies show that 11C-choline PET/CT is highly sensitive in detecting primary tumors and metastases, as well as CT occult metastases, and can potentially provide valuable prognostic information in preoperative staging.

Additional novel PET tracers are being studied to improve the diagnostic accuracy of staging urothelial cancer. Examples include a pilot clinical trial at Thomas Jefferson University studying copper CU-64 TP3805 in patients with urothelial bladder cancer [27] and a study at Hadassah-Hebrew University Medical Center comparing the radiotracer uptake of 11C-acetate to the radiotracer uptake of 11C-choline in patients with urothelial carcinoma of the bladder [28]. Further research must be performed to determine whether these radiotracers can be useful in the staging of UTUC.

## *3.* AI and Multiomics in the Classification and Prognostication of Upper Urothelial Tract Urothelial Carcinoma

Machine learning and multiomics can be useful tools in the classification and prognostication of upper urothelial tract carcinomas. Although no algorithms are routinely used in the clinical setting at this time, work has been carried out to predict the staging and grading of UTUC using deep learning, predict protein-based UTUC subtypes based on hematoxylin and eosin (H&E) slides, and predict overall survival based on inflammatory markers [29,30,31]. 

He et al. used a dataset of 884 patients with UTUC who underwent radical nephroureterectomy and collected clinical data including past medical history and laboratory tests, along with data derived from radiologic imaging, including the presence of hydronephrosis and the longest diameter of the tumor. Their primary prediction endpoints were T-staging and grading based on both 1973 and 2004 WHO Classifications. They trained five different neural network architectures and achieved maximum AUCs of 0.76, 0.804, and 0.824 for T-staging, 1973 grading, and 2004 grading, respectively. 

Another group directly used H&E slides to predict the immunohistochemical expression of UTUC. Using 163 samples, a RESNET50 model was trained to predict the underlying expression of relevant biomarkers. Their model achieved an AUC of 0.62–0.99 (95% confidence interval) and presents a potentially useful tool in guiding therapeutic options for UTUC. 

Lastly, Liu et al. developed a prognostic model for survival, using five inflammatory markers from 483 patients with UTUC. These five markers included neutrophil-to-lymphocyte ratio, monocyte-to-lymphocyte ratio, platelet-to-lymphocyte ratio, systemic immune inflammation index, and systemic inflammation response index. After computing a “systemic immune inflammation score” based on these markers, the authors used random forest and Cox regression models to achieve AUCs of 0.872 and 0.801, respectively, for predicting overall survival at 5 years. 

There has been promising recent work applying deep learning and multiomics methods to UTUC. Future work may incorporate the direct use of radiologic imaging in these machine learning models for UTUC detection, subtyping, and prognostication, as has been performed in abundance for urothelial cancer of the bladder [32,33].

## 4. Image-Guided Percutaneous Biopsy

Percutaneous biopsy of UTUS is rarely performed due to the perceived risk of biopsy tract tumor seeding carried over from case reports of tumor tract seeding for percutaneous biopsy of renal cell carcinoma. While there are a few case reports of UTUC biopsy tract tumor seeding, Huang et al. have shown that percutaneous-image-guided biopsy can be performed safely with no additional risk [34,35]. Thus, in patients for which ureteroscopic biopsy cannot be performed, percutaneous-image-guided biopsy remains a safe option for tissue diagnosis [35]. Multidisciplinary discussion of the risks and benefits of the procedure is recommended prior to percutaneous biopsy.

## 5. Surveillance

After treatment of UTUC, follow-up is recommended to evaluate for recurrent tumor or local/distant metastatic disease. Current European Association of Urology (EAU) guidelines suggest different follow-up imaging pathways depending on whether the primary tumor is high- versus low-risk and if the tumor was treated using radical nephroureterectomy (RNU) or kidney sparing management or partial ureteral resection [1]. Of note, the risk for the development of metachronous bladder tumors is higher than that of metachronous UTUC tumors [6]. Roughly 40% of patients with UTUC go on to develop lower tract UC [8]. This risk decreases 4 years after RNU [6]. 

In low-risk primary UTUC following RNU, CTU is management or partial ureteral resection, postoperative CTU is recommended at 3 months, 6 months, and then yearly for 5 years [6].

In high-risk primary UTUC, postoperative CTU is recommended every 6 months for 2 years and then yearly following RNU. Following kidney sparing management or partial ureteral resection, postoperative CTU is recommended at 3 months, 6 months, and then yearly.

These guidelines generally have weak strength ratings, however, and more data are needed to increase the effectiveness of surveillance guidelines [6].

## 6. Discussion

Multiple imaging modalities are available to the referring clinician for detection of UTUC. CTU demonstrates the highest sensitivity and specificity for evaluation of hematuria including those caused by UTUC. Many factors influence an institution’s choice of single bolus or split bolus technique. Both techniques have similar sensitivity and specificity, each with their own drawbacks: the single bolus technique increases the radiation dose, and the split bolus technique has a higher chance of suboptimal opacification of the distal collecting system. PO or IV hydration prior to imaging is generally favorable, whereas prone imaging or an abdominal belt has not shown to be useful. In younger patients with hematuria with low risk of UTUC, the single bolus technique is likely best as the chances of having to repeat imaging is low. In high-risk patients or those undergoing surveillance, a split bolus technique will likely reduce the patients’ lifetime radiation dose. Additionally, a single bolus technique using DECT will likely provide the best results with reasonable radiation dose. As machine learning algorithms are developed using pathological specimens, improved detection and prognostication are likely possible with first imaging.

In patients unable to undergo CTU, MRU remains a promising alternative, with decreased but still high sensitivity and specificity. Given that the potential treatment plan of UTUC is based on staging, while there are studies evaluating the role of imaging in differentiating between T1 and T2 disease, this has not yet translated to clinical practice. With further trials or through the development of new sequencing and the development of scoring systems similar to VIRADS, MRU sensitivity and specificity may approach that of CTU [36].

While demonstrating lower sensitivity and specificity, CEUS remains a promising modality in detection due to its lack of ionizing radiation. CEUS could be used for contrast evaluation of the kidneys in the setting of hematuria in patients who may not be able to receive contrast.

PET/CT, while currently limited, shows a promising future in its detection role through the development of new radiotracers with very delayed urinary excretion such as 11C-Choline, although radiation dose will still likely remain a limiting factor.

With the future development of algorithms and deep machine learning, it may be possible to obtain diagnostic imaging of equal sensitivity and specificity with single-phase contrast CT or noncontrast MRI. 

## 7. Conclusions

While UTUC is a rare disease, the proper detection and staging of tumor burden are crucial in treatment planning. In clinical practice, CTU has been shown to be the most effective and widely available imaging modality available for the detection of UTUC and all-cause microscopic hematuria. MRU remains an effective detection modality at slightly decreased sensitivity and specificity. Advancements in imaging techniques and artificial intelligence continue to offer a promising future in the detection of UTUC with decreased radiation dose.

## Figures and Tables

**Figure 1 cancers-15-05040-f001:**
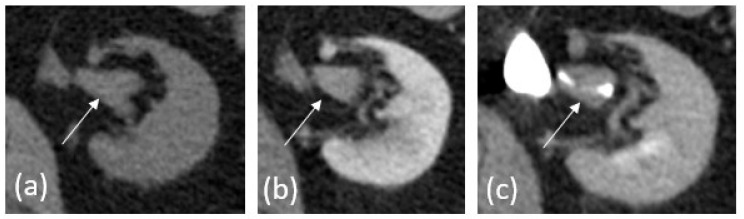
An 81-year-old male with history of bladder UC presenting for surveillance. (**a**) Pre-contrast images demonstrate no significant abnormality. (**b**) Corticomedullary phase demonstrates unequivocal enhancement in tumor (measuring 75 HU compared to 30 HU on pre-contrast images). (**c**) Excretory phase reveals the tumor as a filling defect, outlined by excreted iodinated contrast.

**Figure 2 cancers-15-05040-f002:**
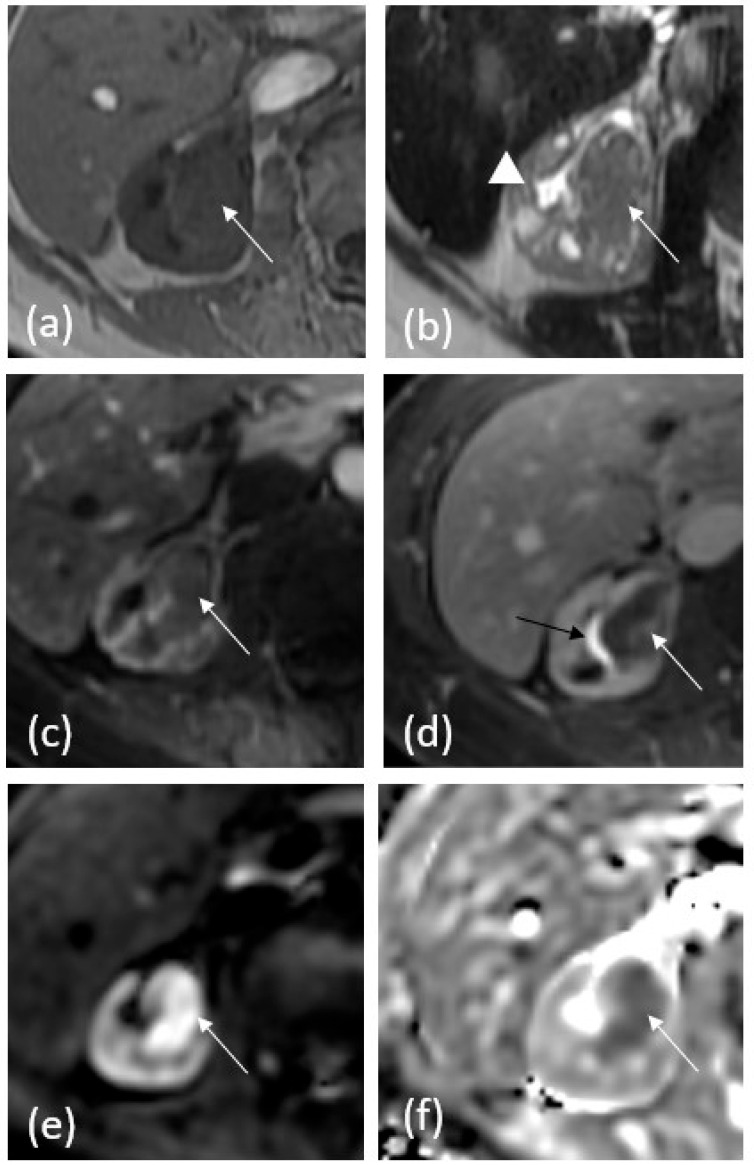
A 64-year-old female with history of right renal mass. (**a**) T1 pre-contrast demonstrates isoechoic mass (white arrow) in the renal pelvis with obliteration of multiple calyces. (**b**) T2-weighted image demonstrates iso- to hypoechoic mass signal again with obliteration of normal calyces and mild associated hydronephrosis (white arrowhead). (**c**) Fat-saturated T1 post-contrast corticomedullary phase demonstrates subtle heterogeneous enhancement within the mass. (**d**) Fat-saturated T1 post-contrast 5 min delayed phase delineates the tumor better, which is enhanced less than adjacent renal parenchyma is. Excreted contrast is noted in the collecting system (black arrow). (**e**) Diffusion-weighted images show increased signal within the mass. (**f**) ADC map shows dark signal in the mass, confirming restricted diffusion.

**Figure 3 cancers-15-05040-f003:**
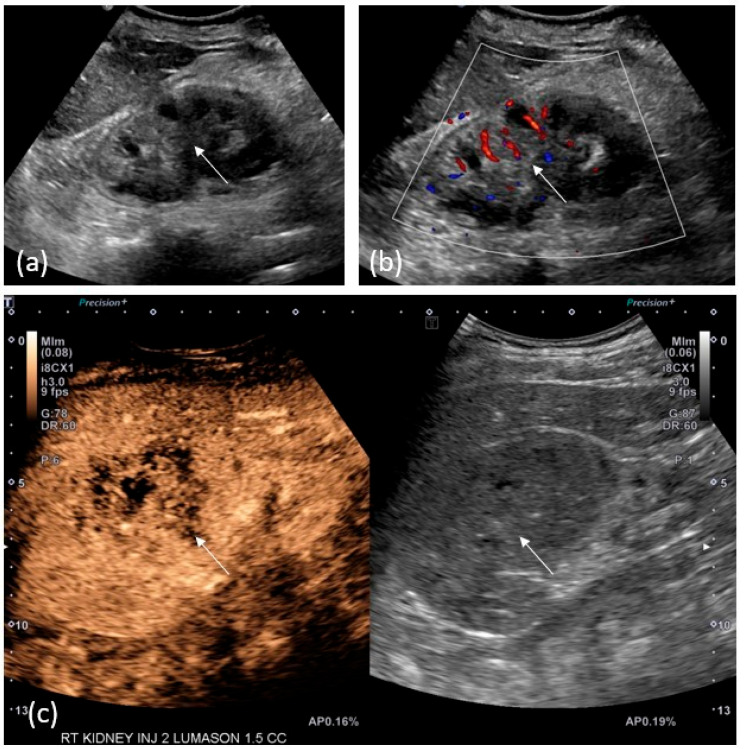
A 71-year-old-woman with right renal mass. (**a**) Grey-scale ultrasound image shows a mass-like hyperechoic area involving the renal pelvis. (**b**) Color Doppler reveals only mild vascularity. (**c**) After injection of 1.5 cc of Lumason, post-contrast images confirm a hypoenhancing mass in that location.

**Figure 4 cancers-15-05040-f004:**
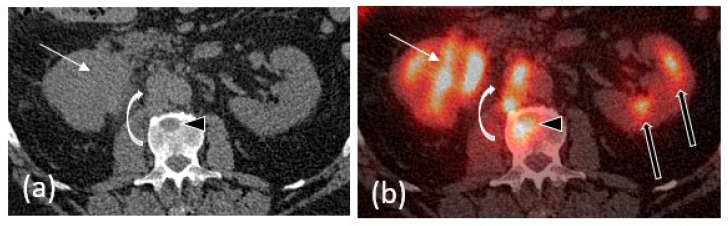
A 63-year-old male with metastatic UTUC. (**a**) Non-contrast CT images demonstrate an infiltrative right renal mass (white arrow) with obliteration of the renal pelvis. (**b**) Fusion PET/CT images demonstrate intense FDG uptake in the mass. FDG avid retroperitoneal lymph node metastases (curved white arrow) and osseous metastases (black arrowhead) are also included in this image. Note the physiologic radiotracer activity in the left renal collecting system (black arrows).

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
