# Peer review of "Imaging in Upper Tract Urothelial Carcinoma: A Review"

_cancers, 2023, doi:10.3390/cancers15205040_

Round 1

Reviewer 1 Report

Congratulations to the authors for the submission entitled "Imaging in Upper Tract Urothelial Carcinoma: A review”.

The review is, in my opinion, clear, comprehensive,and of relevance to the field. No similar reviews have been recently published to my knowledge (pubmed search).

A few references mentioned in the text are not listed in the reference section, and also 2 references listed are not mentioned in the text (see later for details). 

A “conclusions” section at the end of the manuscript  is missed.

Figures are appropriate, although some changes are needed in fig 4 (see later for details)

Please review and correct the following:

Line 8: please delete or either substitute by "urinary"

Lines 18 and 19: calyces are missing (unless they are included within "renal pelvis").

Line 41: Please delete. Some non-containing renal calculi are also seen with noncontrast CT.

Lines 149 and 150: Rewrite the sentence for a better comprehension (i.e. filling defects within the renal pelvis, ureters, and bladder, or pelvicaliceal truncation…)

Line 249: Pierorazio et al are not in the list or references

Line 308: (Slywotzky & Maya, 1994) (Huang, et al., 2015) are not in the reference list

Line 359: Fig 4 a and b legend do not correspond to what is shown in the figures (arrows and arrowheads)

Line 429: reference not mentioned in the text

Line 446: reference not mentioned in the text

Author Response

The review is, in my opinion, clear, comprehensive, and of relevance to the field. No similar reviews have been recently published to my knowledge (pubmed search).

A few references mentioned in the text are not listed in the reference section, and also 2 references listed are not mentioned in the text (see later for details). 

Response: Thank you for bringing this to our attention. The bibliography been updated.

A “conclusions” section at the end of the manuscript is missed.

Response: Thank you for your suggestion. We have included not only a “Conclusions” section, but a “Discussion” section as well.

Figures are appropriate, although some changes are needed in fig 4 (see later for details)

Please review and correct the following:

Response: The figures have been updated. Thank you.

Line 8: please delete or either substitute by "urinary" – changed to urinary

Response: Thanks for bringing this to our attention. This has been changed appropriately.

Lines 18 and 19: calyces are missing (unless they are included within "renal pelvis").

Response: Thank you. We have changed all instances of “renal pelvis” to “pelvicalyceal system” and hope this is unambiguous.

Line 41: Please delete. Some non-containing renal calculi are also seen with noncontrast CT.

Response: We have changed calcified renal calculi to hyperattenuating renal calculi and hope this is more comprehensible.

Lines 149 and 150: Rewrite the sentence for a better comprehension (i.e. filling defects within the renal pelvis, ureters, and bladder, or pelvicaliceal truncation…)

Response: Thank you for your suggestion. This has been changed to “T2 weighted sequences are performed at standard and very long echo times allow for quick hydrographic images. This sequence allows for detection of filling defects or truncation of the pelvicalyceal system, ureters, and bladder, without the need for IV contrast administration (5) (16).” we hope this is more comprehensible for the reader.

Line 249: Pierorazio et al are not in the list or references.

Response: Thank you highlighting this inadvertent error. The reference has been corrected.

Line 308: (Slywotzky & Maya, 1994) (Huang, et al., 2015) are not in the reference list.

Response: The references have been updated.

Line 359: Fig 4 a and b legend do not correspond to what is shown in the figures (arrows and arrowheads).

Response: Thank you for bringing this to our attention. The annotations for Figure 4 have been corrected.

Line 429: reference not mentioned in the text.

Line 446: reference not mentioned in the text.

Response: The references list was out of date and has been updated. Thank you.

Reviewer 2 Report

Dear authors,

Thank you for let me revise your manuscript. As you know UTUC represent a clinical challenge even today and imaging plays a key role in diagnostic. Below you will find a list of suggestion to your work. I would recommend to accept the paper after minor revision.

Here is a list of the revision I suggest you:

  • the way you quote the article in the text is not correct, please refer to the authors Guidelines and fix this 
  • In the DECT paragraph you quote 2 article multiple times, I suggest to quote them only one time at the end (and maybe consider to quote only one of them, considering that they are both review, both from the same group and both of 2023…). Also, are these the only two article about DECT? I made a small research on pubmed and didn’t find much, you should probably say this.
  • Table 1 may be removed, since the majority of readers are familiar with TNM of UTUC. 
  • AI and Machine learning: the studies on H&E and inflammatory markers are out of topic, they should be shortened since the connect with the end of the paragraph
  • Needle tract seeding is one of the fear of the urologist, there are few example on renal cancer, but as you said I didn’t find anything on UTUC (

      10.1080/21681805.2020.1736149

    )

Author Response

Thank you for letting me revise your manuscript. As you know UTUC represent a clinical challenge even today and imaging plays a key role in diagnostic. Below you will find a list of suggestion to your work. I would recommend to accept the paper after minor revision.

Here is a list of the revision I suggest you:

  • the way you quote the article in the text is not correct, please refer to the authors Guidelines and fix this.
    • Response: We have updated the references section and changed to the Vancouver style.
  • In the DECT paragraph you quote 2 article multiple times, I suggest to quote them only one time at the end (and maybe consider to quote only one of them, considering that they are both review, both from the same group and both of 2023…). Also, are these the only two article about DECT? I made a small research on pubmed and didn’t find much, you should probably say this.
    • Response: Thank you for the suggestion. We have mentioned that little literature is currently available and have cited the sources only once at the end.
  • Table 1 may be removed since the majority of readers are familiar with TNM of UTUC.
    • Response: Thank you for your suggestion. We have removed Table 1.
  • AI and Machine learning: the studies on H&E and inflammatory markers are out of topic, they should be shortened since the connect with the end of the paragraph.
    • Response: Thank you. We understand much of the details used are out of scope for the article. We have removed names of the specific tumor markers used for machine learning algorithms.
  • Needle tract seeding is one of the fear of the urologist, there are few example on renal cancer, but as you said I didn’t find anything on UTUC (1080/21681805.2020.1736149
    • Response: We have added the following to the beginning of the image guided percutaneous biopsy section. “Percutaneous biopsy of UTUS is rarely done due to the perceived risk of biopsy tract tumor seeding carried over from case reports of tumor tract seeding for percutaneous biopsy of renal cell carcinoma.”

Reviewer 3 Report

General comment

The manuscript entitled “Imaging in Upper Tract Urothelial Carcinoma: A review” by Tsikitas, Hopstone and Duddalwar, aims to review the strengths and the weaknesses of different imaging techniques available in the common clinical practice regarding UTUC. Despite the interesting topic, the manuscript requires more than a few corrections in order to be suitable for publication. Nevertheless, I’m positive that the authors could efficiently improve their paper according to a few major an minor suggestions.

Firstly, after the abstract, which is almost too generic, an introductive paragraph would be required, despite the presence of the “detection and diagnosis” paragraph which could be assimilated to this role. In particular, after reporting the epidemiological data of this relatively rare cancer, few clinical data about diagnosis and potential treatment (which is influenced by location, numerosity of lesions and lymph node status) should be reported. Additionally, the role of a proper and timely detection should be also reported. Also see DOI: 10.1016/j.clgc.2023.08.001

The paragraph about UTUC stage should be reported before the description of imaging techniques.

If possible, try to be homogeneous with different imaging techniques in terms of construction of the paragraph and the number of words. A proper build could be reporting firstly data regarding sensitivity and specificity then reporting studies and pros and cons.

The role of US/CEUS could be further assessed considering the lack of ionizing radiation and the limited cost compared to other techniques – despite all the limitations.

The percutaneous biopsy is almost never done in the common clinical practice and should be reported.

Two other paragraphs should be added: a discussion, summarizing your point of view on the different imaging techniques, the role of novel imaging methods (as well as the role of radiomics and artificial intelligence) and future perspectives and a proper conclusive paragraph.

minor typos and grammar check

Author Response

The manuscript entitled “Imaging in Upper Tract Urothelial Carcinoma: A review” by Tsikitas, Hopstone and Duddalwar, aims to review the strengths and the weaknesses of different imaging techniques available in the common clinical practice regarding UTUC. Despite the interesting topic, the manuscript requires more than a few corrections in order to be suitable for publication. Nevertheless, I’m positive that the authors could efficiently improve their paper according to a few major an minor suggestions.

Firstly, after the abstract, which is almost too generic, an introductive paragraph would be required, despite the presence of the “detection and diagnosis” paragraph which could be assimilated to this role. In particular, after reporting the epidemiological data of this relatively rare cancer, few clinical data about diagnosis and potential treatment (which is influenced by location, numerosity of lesions and lymph node status) should be reported. Additionally, the role of a proper and timely detection should be also reported. Also see DOI: 10.1016/j.clgc.2023.08.001. Added and rearranged.

Response: Thank you for your feedback. We have created an introduction paragraph with some of the information taken from the Detection and Diagnosis section and some new information added including clinical data for diagnosis and treatment options and in regards to timely detection with your suggested citation.

The paragraph about UTUC stage should be reported before the description of imaging techniques.

Response: The UTUC staging section has been moved as suggested. Thank you for your feedback.

If possible, try to be homogeneous with different imaging techniques in terms of construction of the paragraph and the number of words. A proper build could be reporting firstly data regarding sensitivity and specificity then reporting studies and pros and cons.

Response: Thanks for your feedback. We have tried to match the formatting better and as suggested using sensitivity/specificity data first, then descriptions of techniques. We have also added a discussion section where we again discuss pros and cons.

The role of US/CEUS could be further assessed considering the lack of ionizing radiation and the limited cost compared to other techniques – despite all the limitations.

Response: We added specifically how US and CEUS are ideal due to their lack of ionizing radiation. Additionally, we suggest a path forward for increasing its use in the setting of all cause hematuria. US and CEUS can be used for evaluation of the kidneys with a single excretory CT phase used to evaluate the ureters and bladder, thus decreasing radiation exposure while maintaining a similar sensitivity and specificity for urothelial carcinoma detection in the pelvicalyceal system, as well as for RCC, and nephrolithiasis.

The percutaneous biopsy is almost never done in the common clinical practice and should be reported.

Response: Thank you. We have added a sentence framing that it is rarely performed due to perceived risk  from biopsy tumor tract seeding of renal cell carcinoma.

Two other paragraphs should be added: a discussion, summarizing your point of view on the different imaging techniques, the role of novel imaging methods (as well as the role of radiomics and artificial intelligence) and future perspectives and a proper conclusive paragraph.

Response: Thank you for bringing this to our attention. We have added a discussion and a conclusion section to our manuscript.

Clinical data about diagnosis and potential treatment should be included. Describe how the potential treatment of UTUC is influenced by its location, numerosity of lesions, and lymph node status.

Response: Thank you. This has been addressed in our detection and diagnosis section, as well as in our discussion section.

Reviewer 4 Report

The authors review imaging in UTUC. This review is comprehensive, thorough, of very high quality, and we pay our respects to the authors. Most of the recent papers included are suitable for updating UTUC's knowledge of imaging. This is a strong recommendation for urologists. Please consider the following points.

 1)It is desirable to be able to differentiate between T1 or less and T2 or more on imaging before surgery in order to decide whether to perform preoperative chemotherapy or lymph node dissection. However, at present, it is difficult to make this distinction, but what do you think? How about adding this point to your UTUC staging session?

Author Response

The authors review imaging in UTUC. This review is comprehensive, thorough, of very high quality, and we pay our respects to the authors. Most of the recent papers included are suitable for updating UTUC's knowledge of imaging. This is a strong recommendation for urologists. Please consider the following points.

1) It is desirable to be able to differentiate between T1 or less and T2 or more on imaging before surgery in order to decide whether to perform preoperative chemotherapy or lymph node dissection. However, at present, it is difficult to make this distinction, but what do you think? How about adding this point to your UTUC staging session?

Response: Thank you for your feedback and suggestion. We agree that differentiating between T1 and T2 disease would be greatly beneficial in treatment planning, however current standard imaging practices are limited in this regard . We suggest in our discussion section investigating whether adding a small field of view sequence in an MRU have better sensitivity to this end.

Round 2

Reviewer 3 Report

The authors improved the manuscript accordingly to previous suggestions.

minor checks